# The potential of resilience indicators to anticipate infectious disease outbreaks, a systematic review and guide

**Clara Delecroix**[1,2]*, **Egbert H. van Nes**[1], **Ingrid A. van de Leemput**[1], **Ronny Rotbarth**[1], **Marten Scheffer**[1], **Quirine ten Bosch**[2]*

**1** Department of Environmental Sciences, Wageningen University, Wageningen, The Netherlands,
**2** Quantitative Veterinary Epidemiology, Wageningen University, Wageningen, The Netherlands

* clara.delecroix@wur.nl (CD); quirine.tenbosch@wur.nl (QTB)

**Data Availability Statement:** All relevant data are within the main text and tables and its Supporting information files.

## Abstract

To reduce the consequences of infectious disease outbreaks, the timely implementation of public health measures is crucial. Currently used early-warning systems are highly context-dependent and require a long phase of model building. A proposed solution to anticipate the onset or termination of an outbreak is the use of **so**-called resilience indicators. These indicators are based on the generic theory of critical slowing down and require only incidence time series. Here we assess the potential for this approach to contribute to outbreak anticipation. We systematically reviewed studies that used resilience indicators to predict outbreaks or terminations of epidemics. We identified 37 studies meeting the inclusion criteria: 21 using simulated data and 16 real-world data. 36 out of 37 studies detected significant signs of critical slowing down before a critical transition (i.e., the onset or end of an outbreak), with a highly variable sensitivity (i.e., the proportion of true positive outbreak warnings) ranging from 0.03 to 1 and a lead time ranging from 10 days to 68 months. Challenges include low resolution and limited length of time series, a too rapid increase in cases, and strong seasonal patterns which may hamper the sensitivity of resilience indicators. Alternative types of data, such as Google searches or social media data, have the potential to improve predictions in some cases. Resilience indicators may be useful when the risk of disease outbreaks is changing gradually. This may happen, for instance, when pathogens become increasingly adapted to an environment or evolve gradually to escape immunity. High-resolution monitoring is needed to reach sufficient sensitivity. If those conditions are met, resilience indicators could help improve the current practice of prediction, facilitating timely outbreak response. We provide a step-by-step guide on the use of resilience indicators in infectious disease epidemiology, and guidance on the relevant situations to use this approach.

## Introduction

Infectious disease outbreaks are a leading cause of mortality worldwide, especially in low-income countries and for children [1], with substantial economic and psychological

**Funding:** This publication is part of the project 'Preparing for Vector-Borne Virus Outbreaks in a Changing World: a One Health Approach' (NWA.1160.1S.210), which is (partly) financed by the Dutch Research Council (NWO).

**Competing interests:** The authors have declared that no competing interests exist.

repercussions. Prevention measures such as vaccination and non-pharmaceutical interventions can reduce the consequences of epidemics, and even eliminate some diseases [2]. Measures are most effective if executed before cases start increasing exponentially. However, as outbreaks are hard to anticipate, control efforts often start too late.

Early warning systems have been developed to predict when and where outbreaks will start [3]. These typically depend on the statistical association between the risk of an outbreak and predictive variables. The development of such methods requires having access to various data sources, testing associations, and building statistical models [4]. Diverse factors can be used as predictors, such as climate, geographical settings, population, or socioeconomic data. The use of early warning systems to anticipate outbreaks and predict their consequences have shown to be effective in some cases, for instance, in the anticipation of malaria as well as influenza outbreaks [5, 6]. Other early-warning systems, such as Google Flu Trends, yielded more modest and variable performance and showed rather low associations between the predictors and the risk of an outbreak [7].

Early-warning systems are highly context-dependent, and no standard protocol to build and evaluate them has been proposed [8]. They require consistent parametrization and model fitting. Moreover, complex interactions between the variables, as well as confounding effects, are hard to capture. Developing such models is a long and fastidious process and requires a long cycle of evaluations and adaptations. Further, previously effective early-warning systems might become outdated due to changing conditions and have to be updated [9]. As such, early-warning systems require regular rounds of re-evaluation. A generic, model-free approach would be valuable to improve and complement outbreak anticipation. The use of resilience indicators could be such a generic approach, and was shown to be effective in detecting critical transitions in other complex systems [10].

The start of an outbreak can be defined as a critical transition, a phenomenon observed in many complex systems. Complex systems are defined as systems involving many components interacting with one another and thus leading to non-linear behaviors that are hard to predict. Examples of complex systems are financial markets, ecosystems, the climate and, indeed, infectious diseases in populations. In complex systems, a critical transition occurs when a small change in an underlying condition brings the system across a critical threshold beyond which change becomes self-propelling, driving the system towards a new state. Many complex systems may undergo critical transitions. For instance, financial markets may collapse [11], vegetated ecosystems may shift to a barren state [12], and coral reefs may be overgrown by macroalgae [13]. Being able to anticipate such shifts could enable to prevent their consequences.

Mathematically it can be shown that systems become slow close to a critical transition. This phenomenon is known as critical slowing down [10]. It implies that approaching a critical transition, systems are expected to lose their resilience, i.e., the ability to maintain their normal stabilizing dynamics (e.g., a disease-free state) when subjected to disturbances [14]. In such situations, they are found to recover more slowly from external perturbations. It is usually not possible to directly measure the recovery rate of a system. Therefore, statistical indicators of critical slowing down (e.g., variance, autocorrelation) are computed from representative time series to estimate how close the system may be to undergoing a critical transition [15]. We will refer to these metrics as resilience indicators. Some more background is provided in Box 1.

Pathogen transmission is a complex dynamic process too, as it involves many individuals interacting with one another. When an epidemic starts, the system undergoes a critical transition from a disease-free state to disease emergence. This happens when the effective reproduction number R, i.e., the number of secondary cases arising from an average infected individual in a population, exceeds one. This can be due to a gradual change in conditions, such as a

## Box 1. Critical slowing down to anticipate sharp changes

When conditions change, some complex systems can approach a critical transition, which is a threshold where they lose their stability. Before the threshold is reached, they lose their resilience which is reflected in the intrinsic properties of the system. In particular, the recovery from perturbations becomes slower; a phenomenon called critical slowing down.

As the slower recovery of the system pushed by external perturbations can often not be measured directly, statistical metrics are used as a proxy. They are referred to as resilience indicators. This loss of resilience can be observed in the time series of the system. Since most systems are constantly affected by external perturbations, the increasing time to return to equilibrium is visible in the autocorrelation structure of the time series [15]. When looking at this structure, significant trends are displayed as the system approaches the transition. A rolling window is used to measure these trends: indicators are calculated repeatedly in overlapping subsets of the data to reveal their evolution over time [17].

Similarly, indicators of complexity can be used to anticipate a critical transition. These indicators measure the complexity of a system, defined as its level of disorder. Similar to resilience indicators, complexity indicators are expected to display trends prior to a critical transition, as the complexity of a system is expected to change when approaching a sharp change. However, complexity measures as an indicator of an upcoming critical transition yielded contrasting results in previous studies in other fields [18, 19].

In general, the critical transition in models of infectious diseases is mathematically a transcritical bifurcation. This means that below the critical threshold $R = 1$, the system represented by the number of cases is stabilized at a disease-free state, where only a few cases are observed. Once the threshold $R = 1$ is crossed, the disease-free state becomes unstable as the disease emerges, and major outbreaks can take place. As the critical threshold $R = 1$ is approached, the system's recovery time increases. This means that, when for instance perturbed by the introduction of infected individuals, the number of cases will take longer to vanish (Fig 1B). When the threshold of $R = 1$ is crossed, the disease-free equilibrium becomes unstable (Fig 1C): any perturbation, i.e., the introduction of an infected individual, can result in a major outbreak.

decrease in vaccination rates or improving climatic conditions for the pathogen. Critical slowing down is expected in epidemiological systems prior to R crossing one [16]. Therefore, resilience indicators could theoretically be used to anticipate epidemiological critical transitions based only on incidence time series, allowing to improve timely decision-making. However, the method raises challenges regarding the quality of data required, the processing of the data, and the data interpretation.

This review summarizes the latest findings on the application of resilience indicators to anticipate disease outbreaks based on simulated and real-world data. We address the types of disease, data types, and types of transition suitable to be anticipated using resilience indicators. We review the sensitivity of resilience indicators in public health contexts and discuss their limitations.

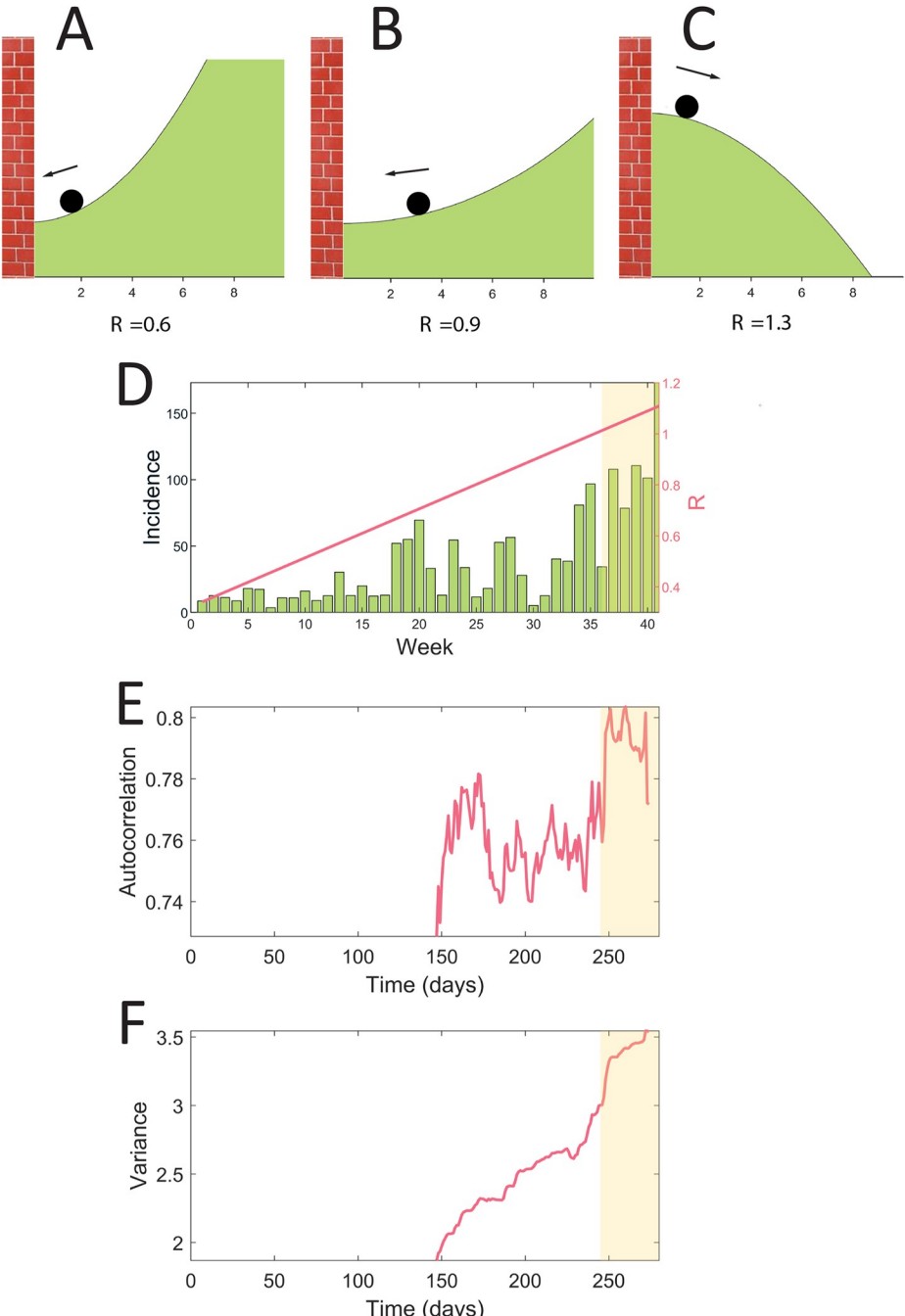

**Fig 1. Illustration of resilience indicators based on simulated data using an SIR model.** In the model, the transmission rate increases linearly over time, resulting in a critical transition when R crosses one. **A-C** are potential landscapes, showing the energy of the system for different states. The ball represents the state of the system. **A** R is relatively far from the threshold: the system will recover easily from an external perturbation. **B** R is close to the critical threshold: the potential to recover from external perturbation is low, and the system undergoes critical slowing down. **C** The threshold is crossed: the system will stabilize at a state for which the disease is endemic. **D** incidence time series generated using a SIR model. The system is undergoing a critical transition: R increases linearly over time until it crosses one (shaded area). **E** and **F** are associated resilience indicators calculated in the simulated time series (daily resolution) using a rolling window. We observe a significant increase of autocorrelation and variance prior to the outbreak.

## Material and methods

We performed a comprehensive literature review to evaluate the current knowledge of resilience indicators to anticipate infectious disease critical transitions. An information retrieval process was performed to review the state of the art of these indicators applied to infectious disease epidemiology. Targeted studies were peer-reviewed research publications using resilience indicators as early warning signals to anticipate infectious disease transitions. The review protocol was not registered.

### Search strategy

This review focuses on resilience indicators based on the theory of critical slowing down. Two high-impact papers published in Nature and Science, cited 2,431 and 1,191 times respectively, are the main references regarding the theory of critical slowing down [10, 20]. We assumed that any study using this theory would cite one of these papers. We carried out a forward citation search intersected with a thematic search to avoid retrieving too many irrelevant results. The search was carried out on September 1st, 2022, using Scopus.

Among the studies citing one of these two papers, a thematic search was performed to only retrieve studies aiming at anticipating critical transitions related to infectious disease outbreaks. The keywords used for the thematic search were *outbreak*, *epidemic*, *disease*, *infecti\**, *ill\**, *epidemiolog\**, *pest*, *virus*, *pandemic*, *bacteria*, *pathogen*, *parasite*. To ascertain that the keywords were relevant, we also checked if adding the name of the top 20 infectious diseases according to WHO in the search keywords would yield new results. This did not result in additional results.

A specific search in the main databases was also used to prevent missing key studies. This additional search also prevented us from missing studies that did not cite one of the two key studies mentioned above. Scopus, Web of Science and PubMed were used for the database search. The search was performed using all keywords of the thematic search described heretofore combined with the term "*early-warning signals*" using "*AND*". The search was purposely kept specific to avoid retrieving too many irrelevant results.

### Selection

The selection was then performed (Fig 2). Pathogens affecting humans or animals were the point of focus. Vegetal or crop pathogens were excluded. Indicators based on the theory of critical slowing down were the point of focus; other methods to anticipate outbreaks were excluded. Only primary publications were considered. The first selection was made based on the title and abstract. We retrieved 60 publications in this round. The second round of selection was based on full-text, using the selection criteria (Table 1), retrieving a final 37 publications (Fig 2, S3 Table). Both rounds of selection were performed by two reviewers independently (CD, RR). At the end of each round, disagreements were discussed with both reviewers until a consensus was reached.

### Classification

We classified the included studies based on the following criteria (S2 Table):

- The type of disease studied: generic disease, seasonal disease, vector-borne disease, or COVID-19.

- Identified best performing indicator: the indicator yielding the best performance to anticipate disease transition in the study.

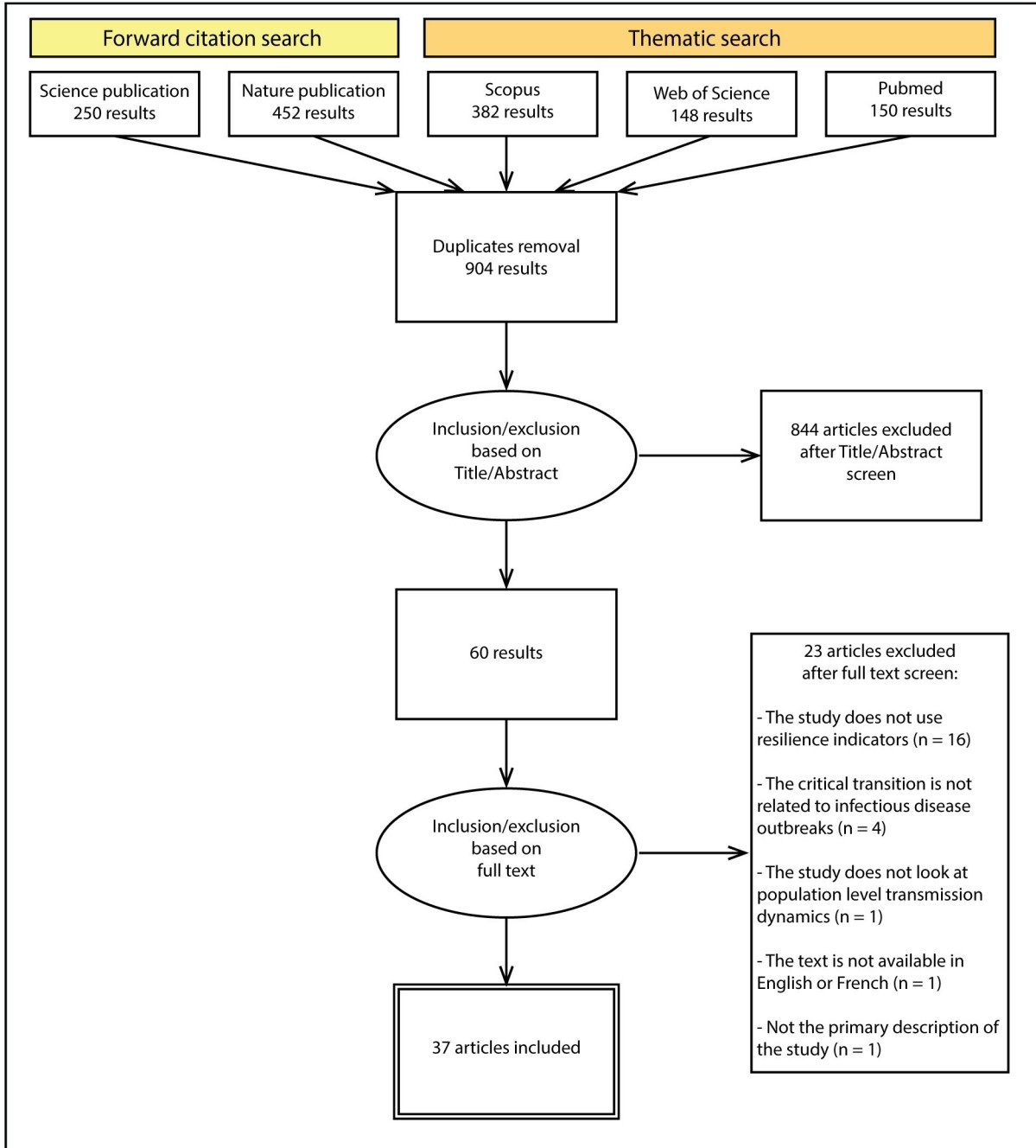

**Fig 2. PRISMA flowchart.** PRISMA flowchart of the literature search process.

- The type of data used: simulated using mechanistic models or real-world data.

- The type of transition anticipated: onset of an outbreak or termination/elimination.

The following data were extracted from the included studies: authors, year of publication, research question, performance of the indicators (quantification of how often indicators could anticipate an upcoming transition, as reported in the study), and method to estimate the performance, false positive rate and lead time (how long in advance the transition could be

**Table 1. Inclusion and exclusion criteria.**

| Inclusion criteria | Exclusion criteria |
|---|---|
| 1. The study is about the use of resilience indicators<br>2. The study investigates critical slowing down<br>3. Resilience indicators are used to forecast a disease critical transition<br>4. The outbreak is caused by an infectious disease affecting humans or other mammals | 1. The study does not use resilience indicators based on the theory of critical slowing down to anticipate the critical transition<br>2. The critical transition is not related to infectious disease outbreaks<br>3. The pathogen does not affect humans or other mammals<br>4. The study does not look at population-level transmission dynamics<br>5. The text is not available in English or French<br>6. Not the primary description of the study. |

anticipated). Finally, when applicable, we extracted the performance of the two most popular indicators, variance and autocorrelation. The data were extracted by two reviewers independently (CD, IAL, EHN). Disagreements were discussed at the end of the information retrieval process until an agreement was reached. The studies' methodological quality and potential biases were also extracted and discussed in the narrative synthesis.

## Results

Among the retrieved studies, 37 met the inclusion criteria. Included studies were published between 2013 and 2022. There has been an increasing interest in resilience indicators to anticipate disease outbreaks, and an increasing number of studies have been published on that topic, especially since 2020 when COVID-19 data became publicly available (Fig 3A). Many of the studies (n = 15, 41%) did not focus on a specific disease and used generic models of infectious diseases to investigate critical slowing down. In the studies investigating specific diseases (n = 22, 59%), 12 different diseases were studied, the main one being COVID-19 (n = 9, 24%). A total of 20 indicators were investigated, the most popular being variance, autocorrelation, mean, and coefficient of variation. These indicators were reported to be among the best-performing ones, respectively in 51% (n = 19), 32% (n = 12), 22% (n = 8) and 22% (n = 8) of the studies. Most of the time, resilience indicators were calculated in simulated data only to anticipate factitious critical transitions (n = 20, 54%). However, the performance of resilience indicators was also investigated on real-world data in a few studies (n = 17, 46%) (Fig 3B). The onset of outbreaks was most often examined (n = 32, 86%). The termination of outbreaks was investigated in a few studies (n = 10, 27%) (Fig 3C). When quantified, the performance was typically calculated using the area under the ROC curve (AUC). We will further refer to the AUC by (prediction) performance, unless specified otherwise.

### Indicators of resilience and complexity

A large variety of indicators can be used to monitor resilience. In the included studies, 20 different indicators were investigated in total (S1 Table). In n = 24 studies (65%), the reported best-performing indicators were autocorrelation, variance, mean, or coefficient of variation. Other well-performing indicators were the logarithmic distance, and composite indicators. The best-performing indicator may vary by disease system (S1 Table). For example, wavelet reddening provided the best performance with periodic data [21], whereas the coefficient of variation outperformed other indicators in anticipating immune-waning induced re-emergence of a disease [22]. Here, we describe the use of variance and autocorrelation as well as alternative indicators such as combinations of indicators, dynamical network markers, and deep learning algorithms.

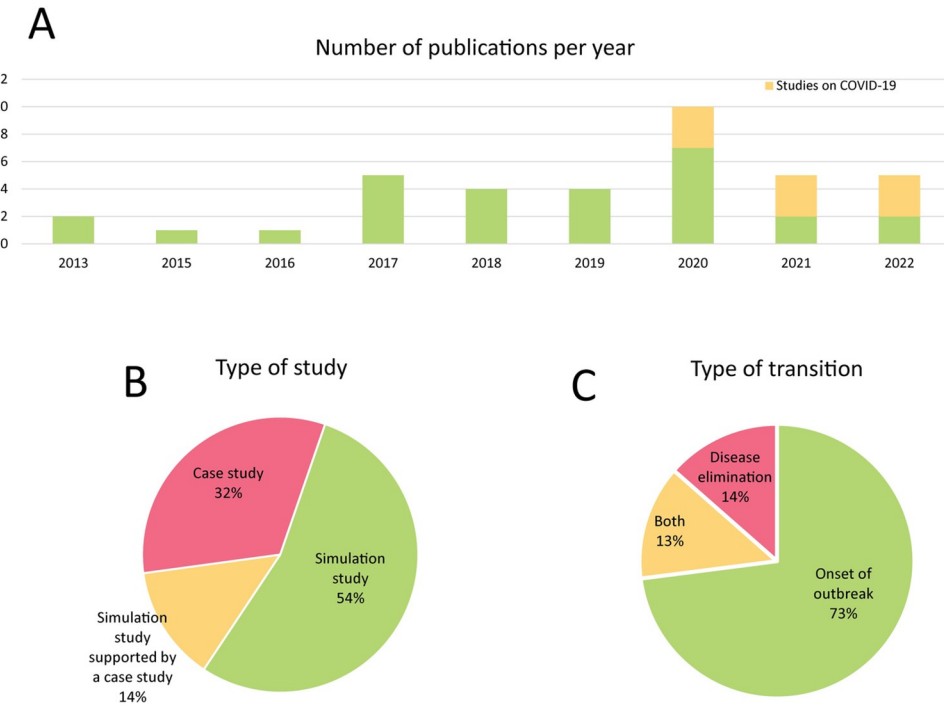

**Fig 3. Overview of the 37 papers included in this review. A** Number of included papers per year. The number of studies on resilience indicators to anticipate epidemics has shown an increasing trend in the last few years. Since 2020, more studies have been published as data on the COVID-19 pandemic became publicly available. **B** Included papers classified according to the type of study into three categories: case studies, simulation studies, and simulation studies supported by case studies. **C** Included papers classified according to the type of transition, into three categories: the onset of an outbreak, disease elimination, and both.

Variance was reported to be one of the best indicators in 19 studies (51%), yielding a prediction performance between 0.5 and 1. However, it is not robust to all types of transition and stochasticity. Supporting Dakos et al.'s findings [23], O'Regan et al. found that variance displays a different trend depending on the type of data, the type of transition and the type of stochasticity [24]. O'Regan et al. showed that specific types of noise could alter the trend in variance: a decrease or no trend at all was sometimes observed, making variance an unreliable indicator in those cases [24].

Autocorrelation, coefficient of variation, and power spectrum are more robust to the type of stochasticity compared to variance: an increase is expected prior to a critical transition. Additionally, autocorrelation is reported to be the best-performing indicator in n = 12 (32%) studies, is robust to data imperfections (section "Data imperfections"), and yielded a performance ranging from 0.2 to 1.

Combinations of indicators have also been studied to anticipate disease emergence. Brett et al. used a supervised learning algorithm to establish an optimal weighted combination of indicators, including mainly skewness, kurtosis, and coefficient of variation [25]. The performance of this combination of indicators was investigated in simulated as well as real-world data. The authors yielded a prediction performance between 0.7 and 0.85 in anticipating several diseases' re-emergence, such as mumps and pertussis. The lead time, namely how long in advance an upcoming outbreak is detected, was between 6 months and 4 years. Similarly, O'Brien et al. could anticipate 2 of the 3 COVID-19 waves in the UK with a lead time ranging from 0 to 48 days using a composite of variance, autocorrelation and skewness [26].

When case reports are discriminated between locations, dynamical network markers (DNM) can be used to anticipate disease (re-)emergence. These indicators were investigated in 5 studies [27–31]. Locations were integrated into a weighted network structure using information on transport between these regions, traffic conditions and population. The correlation of the number of cases between the locations was used to calculate the landscape network entropy index [27, 30, 31], or the minimum spanning tree [28, 29]. The sensitivity of this method ranged from 0.74 to 1, with a lead time between 3 days to 2 months [27–31].

Apart from resilience indicators, some studies investigated the performance of indicators of complexity to anticipate critical transitions [32–34]. Complexity indicators measure the system's level of disorder. In the included studies, six indicators of complexity were investigated: Fisher information [32], Kolmogorov complexity and Shannon entropy [34], mutual information, joint counts, and Geary's C coefficient [33] (S1 Table). In accordance with previous studies, complexity indicators had a lower performance than resilience indicators [18, 19], and failed to identify a transition in one study [32].

Lastly, Bury et al. compared the performance of resilience indicators such as variance and autocorrelation to a deep learning algorithm [35]. They found that resilience indicators slightly outperformed their deep learning algorithm in predicting the onset of an outbreak in simulated data (performance of 0.54 for the deep learning algorithm, and 0.55–0.57 for resilience indicators), a result consistent with other included studies [27–29].

## Simulated data

In total, 25 studies used simulated data (68%) to test whether epidemiological systems display signs of critical slowing down, including 20 relying on simulated data only without accompanying a case study (54%). The data were simulated using compartmental SIR-type models. In such models, the population is divided into categories such as susceptible (S), infected (I), or recovered (R) based on their epidemiological status. Individuals transition from one compartment to another. Such models can be kept purposefully generic or be parametrized for a specific disease. Generic models were investigated in 15 studies as a proof of principle for resilience indicators applied to epidemiological systems as well as to investigate additional complexities (further discussed in the section "Dealing with complexities") [16, 21, 24, 33–45]. O'Regan et al. were the first to demonstrate that critical slowing down arises when an epidemic threshold is being approached [38]. These findings were confirmed in more complex epidemiological systems by including vaccination [42], seasonality [21], age structure [34], mosquito-borne transmission [46], or social behavior [37]. Ten simulation studies used compartmental models parametrized for a specific disease, including COVID-19 [47], measles [22, 48–51], pertussis [50], and smallpox [49]. Various mechanisms of (re-)emergence were studied within these studies, such as annual seasonal outbreaks [48], or re-emergence because of decreasing vaccine uptake [22, 50]. In all studies, signs of critical slowing down were displayed before a critical transition, and they could signal an upcoming outbreak with a highly variable performance, between 0.03 and 1. These studies were used to investigate additional complexities arising in epidemiological systems (further discussed in the section "Dealing with complexities"), or to support a case study and confirm findings from real-world data [22, 25, 48, 52].

## Real-world data

In total, 17 studies used real-world data to study the performance of resilience indicators. Nine diseases were studied: measles [22, 48], mumps [25], pertussis [25, 53], lymphatic filariasis (a parasitic worm disease) [54], plague [25], dengue [25], malaria [55], influenza [27, 29], and COVID-19 [26, 28, 30–32, 52, 56, 57]. We distinguish three different categories of diseases

studied: (i) seasonal diseases with R fluctuating around one, (ii) vector-borne diseases, and (iii) COVID-19.

**Seasonal diseases.** Despite the particularly complex dynamical patterns of seasonal diseases, signs of critical slowing down were detected in six case studies on measles [22, 48], mumps [25], pertussis [25, 53] and influenza outbreaks [27, 29]. In these studies, case reports were used to (i) anticipate long-term re-emergence because of a decline in vaccination or an increase in the infection probability [22, 25], (ii) discriminate locations where epidemics would take place or not [25, 53], and (iii) anticipate annual emergence because of seasonal variations [27, 29, 48]. First, Brett et al. used a combination of indicators to anticipate the long-term re-emergence of mumps and pertussis up to several years in advance [25]. Specifically, a combination of resilience indicators could have anticipated the 2004 national mumps outbreak in England with a lead time of four years [25]. Second, they were able to discriminate localities where an outbreak would occur and localities with low levels of transmission based on local case reports. The authors anticipated pertussis outbreaks in nearly all 37 states that experienced one. However, 30 to 50% of the 12 states that did not experience an outbreak raised a false alarm. Third, Chen et al. and Yang et al. were able to anticipate annual influenza outbreaks in Japan in several areas using case reports per location and a weighted network of the locations to compute dynamical network markers [27, 29]. They yielded a performance of 0.898 with a lead time between 3 and 9 weeks.

**Vector-borne diseases.** The anticipation of vector-borne disease transitions using resilience indicators was shown in three studies investigating dengue in Puerto Rico [25], plague in Madagascar [25], and malaria re-emergence in Kenya [55], and lymphatic filariasis elimination [54]. In these studies, re-emergence was a slow process due to respectively the sequential introduction of serotypes, change of transmission route, or decline in treatment efficacy, and elimination was due to mass drug administration. Brett et al. showed that the outbreaks of DENV-2 and DENV-3 in Puerto Rico could have been anticipated with respective lead times of 18 months and 6 months using a combination of indicators [25]. Further, they illustrated the potential anticipation of the 2017 plague outbreak in Madagascar 30 days before its onset using reports of suspected cases [25]. A recent study suggests that these suspected case reports poorly represented the true extent and temporal evolution of the outbreak [58]. While it is not clear how this has affected the results in [25], it highlights the importance of assuring that data used with resilience indicators are a good representation of the underlying disease dynamics to avoid misleading results. Harris et al. showed that the re-emergence of malaria in Kenya could have been anticipated 6 to 24 months prior to the critical transition using resilience indicators calculated over the hospital case counts [55]. Lastly, signs of critical slowing down were displayed prior to the elimination of lymphatic filariasis, as demonstrated by Michael and Madon [54]. The autocorrelation decreased and served to anticipate the elimination of the disease. Although vector-borne diseases display complex dynamics due to the vector-host interactions, their re-emergence and elimination can be anticipated using resilience indicators calculated in case reports or hospital counts.

**COVID-19.** Three studies attempted to anticipate the first wave of COVID-19, despite the sparse data, and yielded contrasting results [26, 32, 56]. Ma et al. used the Fisher information as a critical slowing down indicator using incidence time series from March 2019 in various countries [32]. The author failed to detect critical slowing down. However, Fisher information is generally considered an indicator of complexity. Complexity measures as an indicator of an upcoming critical transition yielded contrasting results in previous studies, possibly explaining why they failed to anticipate the COVID-19 outbreak [18, 19]. Similarly, O'Brien et al. showed an especially high false-negative rate ($0.62 \pm 0.02$) for the first wave due to the short time series and high variability of the data, consistent with previous results [25, 26, 32]. Only one study by

Kaur et al. [56] succeeded at anticipating the emergence of COVID-19 in 7 out of the 9 countries studied, but with no mention of the lead time and false-positive rate.

During the subsequent COVID-19 waves, consistent testing became the norm in most Western European countries, creating a context of high-quality monitoring ideal for the use of resilience indicators, although still yielding contrasting results. Additionally, information on the geographic location of the cases was available. Six studies investigated the use of resilience indicators to anticipate the waves of COVID-19 [26, 28, 30, 31, 52, 57], including three using dynamical network markers [28, 30, 31]. Overall, signs of critical slowing down were detected with a lead time ranging from 0 days to 2 months, and a performance ranging from 0.04 to 1. Dynamical network markers yielded the highest performance, ranging from 0.825 to 1, but they require location data and the implementation of a location network structure. Additionally, Dablander et al. found that fast successions of elimination and re-emergence hampered the performance of resilience indicators as indicators were sometimes still picking signals of disease elimination before a new wave of COVID-19 [52]. They detected signs of critical slowing down in only 16 out of the 27 countries studied, with some countries raising an alarm for only one of the 10 waves studied.

## Dealing with complexities

Eleven publications discussed the prerequisites for resilience indicators to anticipate accurately critical transitions in infectious diseases [16, 21, 22, 24, 40–43, 50, 52, 57]. These will be discussed in detail in the following.

**Data types.** Most studies discussed up to now have used incidence time series to calculate resilience indicators. These data can be obtained from case reports or hospital case counts. Other data types were also explored and compared: prevalence, rate of incidence as well as alternative sources of data such as Google Trends and Twitter data. By reproducing different types of data using mechanistic models combined with an observation process, O'Dea et al. [41], Brett et al. [40], and Southall et al. [42] showed that prevalence and incidence data portray similar trends in resilience indicators prior to disease emergence. Thereby, a prediction performance around 1 using variance was observed prior to disease emergence in prevalence as well as incidence time series [42]. Similar trends in the variance, as well as a similar prediction performance, were observed in rate of incidence data prior to disease emergence. However, variance can display different trends depending on the data type, making it an unreliable indicator [42]. Additionally, alternative sources of data giving an indirect measure of transmission were also investigated: social media data and Google trend data [22]. Pananos et al. looked at the evolution of the amount of pro-vaccine and anti-vaccine tweets and generated time series to anticipate measles re-emergence [22]. This showed a significant trend in the indicators several years in advance, prior to the re-emergence of measles due to a rising anti-vaccine sentiment.

**Resolution of the data.** The number of data points and the temporal resolution of the time series strongly affect the prediction performance. In case studies, the amount of available data ranged from 10 to 30 years of monthly case reports, being around 120 to 360 data points. O'Dea et al. used simulated datasets to investigate the relationship between data quantity and prediction performance [43]. They showed that the observation period should be much greater than the oscillation period of a seasonal pattern. For instance, for an annual seasonal disease, several years of observation should be available. Moreover, the resolution of the data affects the prediction performance of autocorrelation: equidistant data are necessary for a good estimation of autocorrelation, and the collection interval should be smaller than the infectious period [43].

**Data imperfections.** Epidemiological data is subject to imperfect observations due to misreporting and underreporting, data aggregation and reporting delay making it difficult to

report cases accurately. Brett et al. examined the impact of overdispersion, underreporting and aggregation into periodic reports on the prediction performance using simulated time series [40]. Mean and variance were found to be the least impacted indicators by underreporting and aggregation. Strikingly, their predictive powers were unaffected if the data were not highly overdispersed, meaning displaying a high variability, and if the aggregation period was shorter than the infectious period. Other usually top-performing indicators, such as autocorrelation, performed well for aggregated data but were affected by overdispersion and low reporting probability [40]. Additionally, when reporting rate is increasing together with a varying transmission probability, indicators can struggle to distinguish an increase in transmission probability, leading to an outbreak, from an increase in reporting rate. O'Dea showed that using multiple time series can help confirm that the signal in resilience indicators is the result of an upcoming outbreak and not just a change in transmission probability, and that the second factorial moment is an indicator insensitive to the variation of the reporting probability [41].

**Seasonality.**   Another common characteristic of infectious diseases reflected in epidemiological data is seasonality. Miller et al. simulated time series of infectious diseases subject to seasonal patterns by varying the transmission rate periodically with different levels of amplitude. They found that seasonality does not highly affect the performance of the indicators, as for time series with the highest amplitude of seasonal transmission the performance decreased by 0.02 to 0.07 compared to a sensitivity of 0.85 for non-seasonal simulations. Seasonal detrending did not significantly improve the performance, especially in datasets with low amounts of seasonal fluctuations [21]. Dessavre et al. found that detrending can help improve the accuracy of prediction for some indicators in the case of disease elimination in multiple subpopulations for instance, an argument supported by O'Dea et al. [16, 41].

**Speed of change of R.**   The theory of critical slowing down and the use of resilience indicators to anticipate critical transitions are exclusively dedicated to critical transitions caused by a slow change in an underlying condition. This assumption applies to the anticipation of epidemic transitions as well, as shown by Dablander et al. [52]. The authors showed that overall, the performance of resilience indicators decreases as the speed of change of R increases, meaning that resilience indicators fail at anticipating epidemics emerging too fast. Using simulated data of several waves of COVID-19, they found a performance of variance dropping from around 0.99 to 0.6 as the speed of change increases. Proverbio et al. proposed a method to verify the assumption of a slow increase in R [57]. By using a Bayesian approach, they compute the R over time and measure its speed of change. They consider the assumption to be verified if R reaches one in a period much longer than the serial interval of the disease. Additionally, in the case of multi-wave diseases such as COVID-19, the stabilization time between two waves should be long enough for the epidemics to stabilize in a non-endemic state. Dablander et al. showed that when the time between two waves is too short, resilience indicators fail to anticipate the new outbreak and might pick up signals from the elimination of the previous wave instead [52].

## Discussion–Guidelines on how to use resilience indicators in epidemiology

The advantage of resilience indicators lies in the fact that it is a data-driven, generic method applicable to a wide range of epidemiological systems without the need for frequent recalibration. Simulation studies supported by real-world case studies showed that critical slowing down can indeed be detected prior to disease outbreaks or eliminations, using good quality incidence time series. The 37 studies we reviewed suggest that resilience indicators have the

potential to anticipate outbreaks but yield a highly variable sensitivity. Although the AUC was almost always used to quantify the performance, the false positive rate was poorly documented (only reported in one study). As false positives can result in the implementation of unnecessary interventions or even a precipitated halt of disease elimination strategies, it is important to get a better understanding of how the specificity of resilience indicators is affected by complexities in the data and the disease system. Similarly, lead time was not always quantified in the studies (only reported in ten studies), even though this is a key aspect of disease anticipation. Additionally, our information retrieval is likely subject to publication bias, which may result in an overestimation of the performance of these tools.

To bridge the gap between theory and practice, it is necessary to get a better understanding of the factors affecting the performance of resilience indicators as well as the suite of disease and monitoring systems that are best suited for the use of resilience indicators. Here, we present a step-by-step approach to assess whether a disease and its monitoring system are suitable for the use of resilience indicators. We also suggest how such an early warning system may be set up for the system at hand (Fig 4).

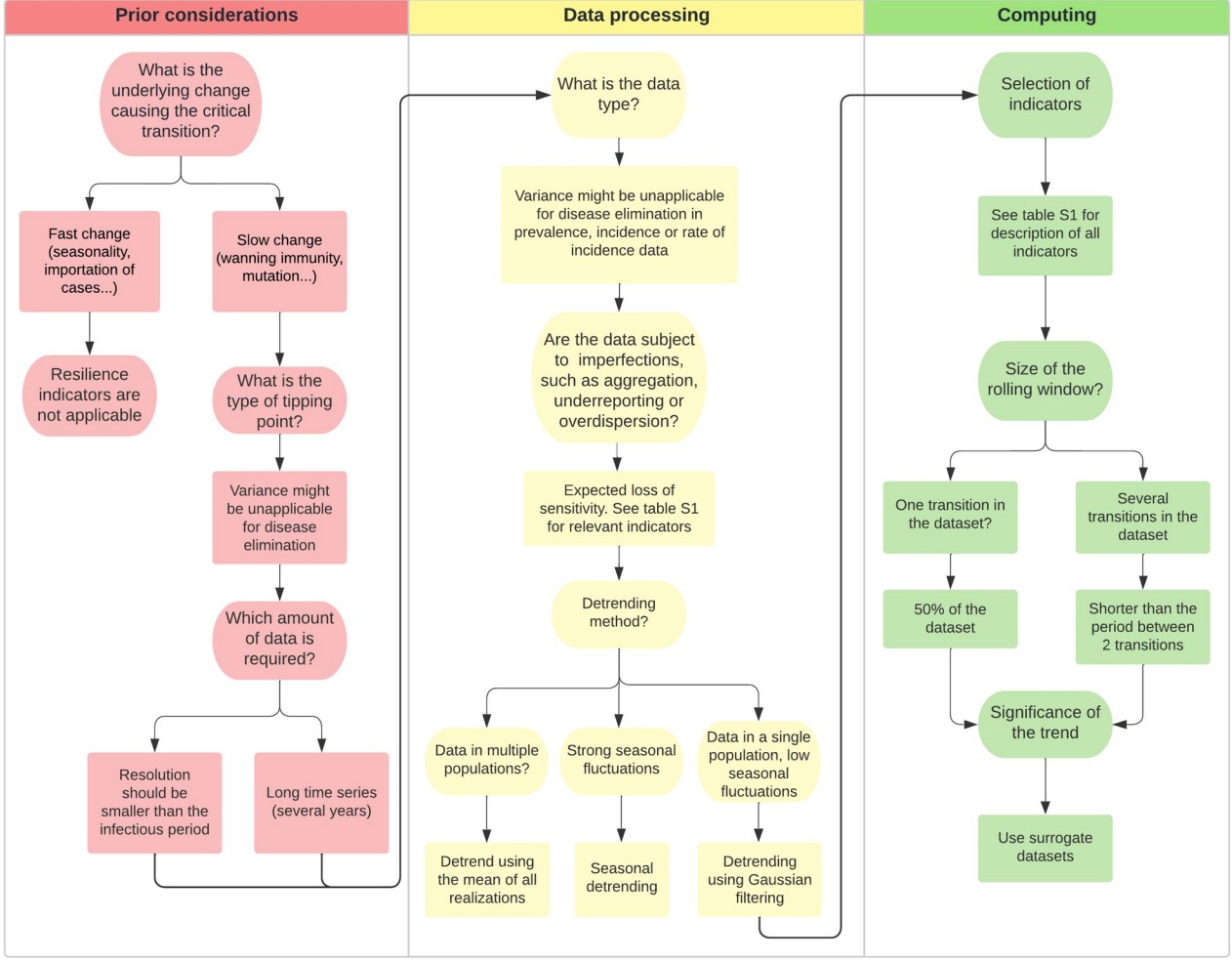

**Fig 4. Decision tree.** Step-by-step approach to use resilience indicators in epidemiology.

### Prior considerations

Although resilience indicators can help anticipate critical transitions, this may only be expected to work in specific contexts. First, we cannot expect signals of critical slowing down prior to a transition in all situations. At least two requirements must be fulfilled to use resilience indicators: suitable data should be available and external conditions should change slowly [15, 52]. We can distinguish several reasons for a new outbreak. A common mechanism is the emergence of a new unknown pathogen due to spillover from wild animals, for example. In this case, no suitable data will be available to observe critical slowing down. Another possibility is a pathogen remaining close to endemicity as their R fluctuates around 1, and that is subject to seasonal variations leading to sudden outbreaks. Under those circumstances, the seasonal change in conditions is likely too fast to detect critical slowing down. By contrast, when the risk for a pathogen to cause an outbreak rises gradually due to changing conditions, the outbreak might be anticipated using critical slowing down. Examples could include changes that may bring the R gradually closer to 1, such as a decline in vaccine uptake, mutation of the pathogen inducing immunity escape, and change in the immunity profile of a population due to waning immunity. Statistical tests have been proposed to check the assumption of slow change [57].

Second, the type of transition can affect the trend in some of the indicators. Disease outbreaks as well as disease elimination can be anticipated using resilience indicators. Prior to both transitions, critical slowing down is displayed in the system, as was shown in simulation studies as well as case studies. However, depending on the type of data, variance might not increase before the elimination of a disease [38]. Thus, autocorrelation should always be the first choice as it displays consistent trends, insensitive to the data types.

Furthermore, enough data points should be available, with a sufficient resolution to capture slowing down in order to anticipate disease critical transitions. The collection interval should be smaller than the infectious period [40] with a reasonable number of data points, meaning that the duration of observation is at least as long as the period of any oscillation in the data [43]. For instance, if the disease is a seasonal disease coming back every winter, at least a year of observations should be available. As a comparison, the included case studies based their analyses on around a decade of monthly case reports. Data should be equidistant for a good estimation of autocorrelation. Additionally, case reports discriminated per location can help improve the prediction performance using dynamical network markers. However, only five studies were published using these types of indicators. Thus, further investigation is required.

Few countries have surveillance systems able to achieve such high-quality data, especially since sufficient data prior to the start of an outbreak is necessary. In general, we can distinguish active surveillance efforts, sentinel surveillance, and passive surveillance. Active surveillance, namely the active seeking of cases of a given disease in a population, allows to estimate the prevalence of a disease with high accuracy but is extremely costly and not achievable in the long term. Sentinel surveillance system, i.e., the monitoring of disease prevalence in a population via a network of general practitioners, can help achieve consistent, good-quality data assuming that these are reported with a sufficient temporal resolution and that the system is sufficiently sensitive [40]. Covid-19 and influenza are examples of diseases that are monitored via sentinel surveillance. Influenza is monitored through sentinel surveillance systems within the GISRS initiative from WHO, and cases are reported with a weekly resolution in the FluID database [59]. Similarly, covid-19 is nowadays consistently monitored via sentinel surveillance systems and sometimes reported in publicly available databases [60–63]. However, the quality of sentinel surveillance systems depends on the health-seeking behavior and access to healthcare facilities in the population. Similarly, the quality of data provided by passive surveillance

systems is context-dependent. Most included case studies relied on data obtained from passive surveillance systems in areas with sufficient access to healthcare facilities and for diseases with sufficient symptomatic cases. However, such systems can underestimate the prevalence in periods of low transmission [64, 65], and hamper the prediction performance of resilience indicators if the reporting rate increases together with the prevalence of the disease [41]. Low-income countries with circulating diseases of poverty such as cholera or Ebola as well as neglected tropical diseases lack a constant surveillance system in place prior to outbreaks and the surveillance is mostly reactive, making the data unsuitable for resilience indicators [66–69].

Alternatively, substitute types of more accessible time series representing the state of an epidemic indirectly could be considered. Critical slowing down in Google trends or social media data was investigated, and significant trends were displayed prior to a measles outbreak [22]. Other types of alternative data could also be envisioned, such as excess mortality data [70], news feed [71] or wastewater surveillance data [72]. Wastewater surveillance, in which biomarkers related to a specific disease are quantified in untreated sewage data, provides real-time data and allows monitoring the state of the epidemic with less effort than by counting the new cases. However, investigations would be required to make sure that critical slowing down is also displayed in this type of data.

## Data processing

Once we know the disease transition is relevant with regard to resilience indicators, pre-processing of the data should be conducted prior to the analysis. Detrending of the data is usually necessary to avoid spurious trends in the indicators due to slow changes in the mean [17]. This is essential, especially for seasonal data. Seasonality affects the spread of a number of diseases, creating periodic fluctuations in the data. These fluctuations have an effect both on variance and autocorrelation, introducing misleading results. When studying a disease subject to periodicity, the number of data points should be much higher than the period. In other terms, if the disease has waves every winter, one should have data over several years. This helps assess if the trend in the indicators is truly due to long-term re-emergence and not to seasonal fluctuations.

Several types of data can represent the state of the system. Incidence time series represent the count of new cases, while prevalence time series count the number of infected individuals at different time points. The rate of incidence is the rate at which newly infected cases occur in a population. The rate of incidence can be estimated from incidence time series using a rolling window approach [42]. Critical slowing down is displayed prior to a transition in all these types of data. However, when using variance as an indicator of resilience, the type of data can affect the trend prior to a transition. Moreover, although the rate of incidence requires additional computations to be obtained, it displayed a more significant trend prior to disease elimination in one study [42]. If dynamical network markers are to be used, it is necessary to build a location network using population data, using the information on transport between these regions, traffic conditions and population.

As monitoring is never perfect, epidemiological data are subject to imperfections. The data are aggregated into weekly or monthly case reports. Underreporting is often observed as a result of asymptomatic cases as well as poor access to health facilities. Moreover, various types of stochasticity are inherent to the data. Again, these characteristics can affect the trend in variance. Furthermore, when combined, imperfections can be detrimental to the performance of resilience indicators. Imperfections likely to be encountered in the data should be clearly stated in order to select relevant indicators. Variance and mean perform poorly when data are highly overdispersed, meaning that data show great variability. Similarly, autocorrelation performs

poorly when the reporting rate is highly overdispersed or when the aggregation period is too high. If the reporting rate is expected to change, the second-order moment could be used as it is insensitive to variations in the reporting rate [41].

### Computing

After pre-processing the data, the resilience indicators can be computed using packages, for instance, in R or Matlab [73, 74].

The indicators should be picked carefully based on the prior reflection presented above. Variance was the top-performing indicator in a majority of studies and least impacted by underreporting and aggregation. However, when choosing variance, the trend can be inverted. Autocorrelation was among the best-performing indicators in a majority of studies, and its trend is not affected by the type of transition. However, the performance of autocorrelation is impacted in the case of low reporting probability and highly overdispersed data, and equally spaced data are necessary to calculate autocorrelation. A variety of indicators can be used for specific situations (S1 Table). Combinations of indicators yielded the best performance [25]. However, the best combinations were determined using an optimization algorithm trained on a large dataset of simulated time series, and their performance remains to be proved in other contexts.

The size of the rolling window should be picked carefully to observe a trend at a consistent scale. An arbitrary value is to take 50% of the size of the dataset as a window size [17]. However, if several transitions occur, then a smaller rolling window size should be picked to be able to observe a trend before each transition. In addition, enough data points should be present in the window in order to accurately estimate the autocorrelation; however, a too-large window will reduce the absolute increase [57]. It is good practice to check the effect of the window size and detrending in a sensitivity analysis [17].

When a trend is observed, its significance needs to be assessed. Due to the sliding window approach, standard statistical tests are not applicable as the observations are not independent. A proposed approach to assess the significance of the trend is to produce surrogate datasets to compare the trend estimates [17]. Several methods to produce consistent surrogate datasets have been proposed and implemented in the resilience indicators packages [73, 74]. The choice of the threshold should be calibrated based on previous data, as a poorly calibrated threshold can induce misleading results [52].

## Conclusion and future directions

To conclude, resilience indicators have the potential to help public health organizations anticipate infectious disease transitions, as they constitute a generic, data-driven method. Real-time calculation of resilience indicators could be put into practice to monitor the risk of an upcoming outbreak, provided sufficient, good-quality case reports are available. However, further investigations are required to strike the right balance between false negative and false positive rates, and lead time. This will differ by setting, disease system, and data availability and quality. To overcome the data and model limitations, a combination with other early-warning systems, as well as other sources of data, might help improve early detection. The potential of such combined approaches remains to be explored. Moving forward, a close collaboration between experts in resilience indicators and public health practitioners is needed to bridge the gap between theory and practice, and determine how and when resilience indicators could contribute to more timely outbreak response.

## Supporting information

**S1 Table. Summary of the indicators and their usage.** Original references refer to primary studies not included in that review that studied the use of the indicators. Mathematical

derivation of the indicators is given in [53].
(XLSX)

**S2 Table. Summary of the included studies.** Summary of the included studies and their classification.
(XLSX)

**S3 Table. Summary of the publications retrieved during database search, and the inclusion/exclusion decisions.**
(XLSX)

**S4 Table. PRISMA checklist.**
(DOCX)

## Author Contributions

**Conceptualization:** Clara Delecroix, Marten Scheffer, Quirine ten Bosch.

**Data curation:** Clara Delecroix, Egbert H. van Nes, Ingrid A. van de Leemput, Ronny Rotbarth.

**Formal analysis:** Clara Delecroix.

**Funding acquisition:** Quirine ten Bosch.

**Methodology:** Clara Delecroix, Egbert H. van Nes, Ingrid A. van de Leemput, Quirine ten Bosch.

**Supervision:** Egbert H. van Nes, Ingrid A. van de Leemput, Marten Scheffer, Quirine ten Bosch.

**Validation:** Clara Delecroix, Ingrid A. van de Leemput, Ronny Rotbarth, Marten Scheffer, Quirine ten Bosch.

**Writing – original draft:** Clara Delecroix.

**Writing – review & editing:** Clara Delecroix, Egbert H. van Nes, Ingrid A. van de Leemput, Ronny Rotbarth, Marten Scheffer, Quirine ten Bosch.

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
