## [Decision Letter · Decision Letter 0]

8 Jun 2023

PGPH-D-23-00589

The potential of resilience indicators to anticipate infectious disease outbreaks, a systematic review and guide

Dear Dr. Delecroix

Thank you for submitting your manuscript to PLOS Global Public Health. After careful consideration, we feel that it has merit but does not fully meet PLOS Global Public Health’s publication criteria as it currently stands. Therefore, we invite you to submit a revised version of the manuscript that addresses the points raised during the review process.

Please submit your revised manuscript by 22/June/2023 If you will need more time than this to complete your revisions, please reply to this message or contact the journal office at globalpubhealth@plos.org. Please include the following items when submitting your revised manuscript:

We look forward to receiving your revised manuscript.

Kind regards,

Reuben Kiggundu

Academic Editor

Journal Requirements:

Additional Editor Comments (if provided):

Reviewers' comments:

Reviewer's Responses to Questions

**Comments to the Author**

1. Does this manuscript meet PLOS Global Public Health’s publication criteria? Is the manuscript technically sound, and do the data support the conclusions? The manuscript must describe methodologically and ethically rigorous research with conclusions that are appropriately drawn based on the data presented.

Reviewer #1: Yes

Reviewer #2: Yes

2. Has the statistical analysis been performed appropriately and rigorously?

Reviewer #1: N/A

Reviewer #2: Yes

3. Have the authors made all data underlying the findings in their manuscript fully available (please refer to the Data Availability Statement at the start of the manuscript PDF file)?

Reviewer #1: Yes

Reviewer #2: Yes

4. Is the manuscript presented in an intelligible fashion and written in standard English?

Reviewer #1: Yes

Reviewer #2: Yes

5. Review Comments to the Author

Reviewer #1: The study is well narrated and elaborated. Currently the issue of infectious disease outbreak needs to be revised and new method should be developed. So that combination approaches should be practiced. So, this study will help in this area. In line 299 re-emergence of malaria is not in all places please specify the country.

Reviewer #2: The manuscript is well written and organized. Data is a correctly available and schemes are clear. It would be interesting to include some evaluation on the different health/surveillance systems of countries, considering the data provided by each system can be a limitation to resilience factors. If ithe evaluation is not possble, I suggest including as a limitation in the article.

6. PLOS authors have the option to publish the peer review history of their article (what does this mean?). If published, this will include your full peer review and any attached files.

**Do you want your identity to be public for this peer review?** For information about this choice, including consent withdrawal, please see our Privacy Policy.

Reviewer #1: No

Reviewer #2: No

---

## [Editor Report · Decision Letter 1]

14 Jul 2023

The potential of resilience indicators to anticipate infectious disease outbreaks, a systematic review and guide

PGPH-D-23-00589R1

Dear Dr. Delecroix,

We are pleased to inform you that your manuscript 'The potential of resilience indicators to anticipate infectious disease outbreaks, a systematic review and guide' has been provisionally accepted for publication in PLOS Global Public Health.

Best regards,

Reuben Kiggundu

Academic Editor
